# Platelet-Rich Plasma Power-Mix Gel (ppm)—An Orthobiologic Optimization Protocol Rich in Growth Factors and Fibrin

**DOI:** 10.3390/gels9070553

**Published:** 2023-07-07

**Authors:** José Fábio Lana, Joseph Purita, Peter Albert Everts, Palmerindo Antonio Tavares De Mendonça Neto, Daniel de Moraes Ferreira Jorge, Tomas Mosaner, Stephany Cares Huber, Gabriel Ohana Marques Azzini, Lucas Furtado da Fonseca, Madhan Jeyaraman, Ignacio Dallo, Gabriel Silva Santos

**Affiliations:** 1OrthoRegen Group, Max-Planck University, Indaiatuba 13343-060, Brazil; josefabiolana@gmail.com (J.F.L.); peter@gulfcoastbiologics.com (P.A.E.); 2PUR-FORM, Boca Raton, FL 33432, USA; jpurita@aol.com; 3Brazilian Institute of Regenerative Medicine (BIRM), Indaiatuba 13334-170, Brazil; palmendonca35@gmail.com (P.A.T.D.M.N.); danfjorge@gmail.com (D.d.M.F.J.); tmosaner@uol.com.br (T.M.); stephany_huber@yahoo.com.br (S.C.H.); drgabriel.azzini@gmail.com (G.O.M.A.); ffonsecalu@gmail.com (L.F.d.F.); 4Department of Orthopaedics, Faculty of Medicine, Sri Lalithambigai Medical College and Hospital, Tamil Nadu 600095, India; madhanjeyaraman@gmail.com; 5SportMe Medical Center, Department of Orthopaedic Surgery and Sports Medicine, Unit of Biological Therapies and MSK Interventionism, 41013 Seville, Spain; doctorignaciodallo@gmail.com

**Keywords:** platelet-rich plasma, platelet-rich fibrin, hyaluronic acid, orthobiologics, osteoarthritis, regenerative medicine

## Abstract

Platelet- and fibrin-rich orthobiologic products, such as autologous platelet concentrates, have been extensively studied and appreciated for their beneficial effects on multiple conditions. Platelet-rich plasma (PRP) and its derivatives, including platelet-rich fibrin (PRF), have demonstrated encouraging outcomes in clinical and laboratory settings, particularly in the treatment of musculoskeletal disorders such as osteoarthritis (OA). Although PRP and PRF have distinct characteristics, they share similar properties. The relative abundance of platelets, peripheral blood cells, and molecular components in these orthobiologic products stimulates numerous biological pathways. These include inflammatory modulation, augmented neovascularization, and the delivery of pro-anabolic stimuli that regulate cell recruitment, proliferation, and differentiation. Furthermore, the fibrinolytic system, which is sometimes overlooked, plays a crucial role in musculoskeletal regenerative medicine by regulating proteolytic activity and promoting the recruitment of inflammatory cells and mesenchymal stem cells (MSCs) in areas of tissue regeneration, such as bone, cartilage, and muscle. PRP acts as a potent signaling agent; however, it diffuses easily, while the fibrin from PRF offers a durable scaffolding effect that promotes cell activity. The combination of fibrin with hyaluronic acid (HA), another well-studied orthobiologic product, has been shown to improve its scaffolding properties, leading to more robust fibrin polymerization. This supports cell survival, attachment, migration, and proliferation. Therefore, the administration of the “power mix” containing HA and autologous PRP + PRF may prove to be a safe and cost-effective approach in regenerative medicine.

## 1. Introduction

Osteoarthritis (OA) is acknowledged as a major degenerative and progressive joint disease responsible for significant pain and disability in the adult population [1]. The incidence of OA across the globe has risen significantly in the last few decades due to metabolic syndrome and aging [2,3,4]. This disease can often be challenging to treat as it presents a multifactorial nature, being mainly characterized by the physiological and architectural changes in the joint compartment as a whole [5]. It is highly influenced by the interplay between innate and extraneous factors, consequently dictating disease progression and the patient’s response to treatments [6]. Based on the Kellgren–Lawrence scale (Table 1), the most common features of OA include progressive loss of articular cartilage, osteophyte formation, thickening of the subchondral bone, escalated synovial inflammation, ligament and meniscal degeneration, and overall joint hypertrophy [3]. The Kellgren–Lawrence scale is an important tool since it has been acknowledged as a valid and reliable radiographic grading system for OA and still remains one of the most widely used in clinical practice and in research [7,8,9].

A variety of modalities have been used to treat osteoarthritis with both pharmacological and nonpharmacological alternatives to avoid surgical intervention [5]. Since OA symptoms are known to be intense, physicians may often prescribe patients a combination of drugs at different stages of the disease, aiming to block inflammatory nociceptive pain [10]. Nonsteroidal anti-inflammatory drugs (NSAIDs), other analgesics, and corticosteroids are commonly prescribed to aid in pain management. However, it is important to emphasize that prolonged administration of NSAIDs is a major health concern [11]. They do, in fact, promote temporary relief, but they can also make the patient more susceptible to the development of serious adverse events such as peptic ulcer disease, acute renal failure, and even myocardial infarction [11]. Nonpharmacological alternatives, on the other hand, are less aggressive in this sense but are still quite limited in terms of regenerative potential [12]. Physical therapy, low-impact exercise, weight loss, physical aids, and nerve ablation are typical examples of nonpharmacological strategies that may assist in the treatment of OA during its initial stages; however, these may not suffice in more severe and advanced stages, such as grade IV OA, where surgical intervention procedures may be inevitable [6,12].

These hurdles have motivated medical experts to develop novel alternative methods that are able to target the pathophysiological progression of OA without promoting secondary problems. Such alternatives are widely known as orthobiologics. However, due to the large number of differing reports in the literature, it is challenging to draw conclusive support regarding their use. In short, these biomaterials are derived from biological components that are naturally found in the body, with the ability to enhance the healing process of orthopedic conditions [13]. 

There are many popular examples given in the literature, such as platelet-rich plasma (PRP), platelet-rich fibrin (PRF), hyaluronic acid (HA), bone marrow aspirate/concentrate (BMA/BMAC), and even adipose tissue [14,15]. These products can have autologous or allogeneic origins and are known to be rich in cellular and molecular components [16]. For decades, these components have demonstrated a significant capacity to modulate not only the pathophysiology of OA but many other disease processes as well, providing great optimism in the field of regenerative medicine [16]. 

Previously, we published a description of two techniques: the BMA Matrix [17] and the Platelet-Rich Plasma Gel Matrix (PRP-GM) [18]. Both techniques consist in the generation of a rich autologous biological matrix; the first one is obtained via the mixture of BMA with HA, and the second one requires mixing PRP, HA, and platelet-poor plasma gel. The objective of this narrative review follows the same principle. We propose a treatment alternative for osteoarthritis, focusing on autologous PRP and PRF, two well-known fibrin-rich products, and their association with HA in order to form a rich biological gel matrix. 

## 2. Methods

The literature was reviewed using PubMed and Google Scholar from January to March 2023 in order to reveal the regenerative medicine potential of fibrin-rich orthobiologic products. The investigation included nomenclature such as the following: “osteoarthritis”, “inflammation”, “orthobiologics”, “fibrin”, “fibrinolytic reactions”, “platelet-rich plasma”, “platelet-rich fibrin”, and “hyaluronic acid”. Only full-text articles in English were considered for review.

## 3. OA Etiopathogenesis

OA progression is driven by the by the intricate interplay of various local, systemic, and external factors, which consequently dictate the progression outcome and the patient’s response to treatment [6]. OA is characterized by multiple hallmarks, such as gradual degeneration of articular cartilage, the formation of osteophytes, thickening of the subchondral bone, increased inflammation of the synovium, deterioration of ligaments and menisci, and an overall increase in joint size [3]. 

The development of OA is the result of a combination of various factors, including genetic predisposition, obesity, traumatic injuries, aging, and the presence of other systemic diseases [19]. It affects the entire joint compartment, and previous research indicates that the degenerative process occurs in two distinct phases. Firstly, in the biosynthetic phase, chondrocytes attempt to repair the damaged extracellular matrix (ECM) repeatedly. Subsequently, in the degradative phase, elevated catabolic enzyme activity promotes digestion of the ECM and inhibits the synthesis of new ECM [10]. In healthy patients, a functional ECM contains a relative abundance of vital organic components, including water, chondroitin sulfate, type II collagen, proteoglycans, HA, and other proteins like fibronectin and laminin [20]. The fibrous components of the ECM are elastin and collagen, which consist of several types of fibrillar collagens such as types I, II, III, V, and XI, as well as nonfibrillar collagens including FACIT types IX, XII, and XIV, short chain types VIII and X, and basement membrane type IV [20]. When balanced, all of these components keep the ECM stabilized, preventing cellular damage.

Continuous biomechanical and biochemical stress causes secondary alterations, resulting in a predominant shift towards catabolic reactions. These biological processes disturb cellular activity, being therefore responsible for cartilage erosion and injury to the subchondral bone and peripheral structures, all of which worsen physical pain and debilitation [21]. The role of the innate immune system in OA has also been suggested by previous research [22]. This system is activated upon recognition of conserved motifs produced by pathogenic agents and/or cellular damage that occurs within tissues [23]. More specifically, this refers to damage-associated molecular patterns (DAMPs), which activate the innate immune system. DAMPs are generated as a result of damage to cellular and cartilage ECM products due to trauma, microtrauma, and even normal aging [24]. Usually, these molecular aggressors are fragments from proteoglycans, proteins, or residual cellular breakdown [25], which trigger a sterile inflammatory response upon interacting with particle recognition receptors (PRR), including toll-like receptors (TLR) on immune cell surfaces, or with cytosolic PRRs, such as nod-like receptors (NLRs) [23,24].

Furthermore, osteoarthritic synoviocytes and chondrocytes produce high levels of MMPs (matrix metalloproteinases) such as MMP-1, MMP-3, MMP-9, and MMP-13 [26]. Synoviocytes secrete proteolytic enzymes and proinflammatory cytokines, including IL-1β, IL-6, and tumor necrosis factor-alpha (TNF-α). These mediators play a role in both the progression of OA and the perception of painful stimuli associated with the condition [27]. Other molecules, such as resistin and osteopontin, are significantly elevated in osteoarthritic synovial tissue, and their increased expression is associated with disease severity [28,29,30]. Although cartilage itself produces most of the catabolic molecules via autocrine and paracrine signaling mechanisms, the synovium has also been reported to produce some chemokines and metalloproteinases that contribute to the degeneration of cartilage [31]. As a result, the breakdown products generated from cartilage degradation, either through biomechanical or biochemical insult, can stimulate the release of collagenase and other hydrolytic enzymes from synovial cells. This process contributes to vascular hyperplasia in the synovial membranes affected by OA [32].

In healthy individuals, articular cartilage is primarily composed of ECM that contains water, collagen, proteoglycans, and a small amount of calcium salt, as well as chondrocytes [33]. The turnover rate of collagen is relatively slow, while the turnover rate of proteoglycan is relatively faster [21]. Chondrocytes are responsible for controlling this process and are responsible for synthesizing molecular components, including proteolytic enzymes, which regulate the breakdown of the ECM in articular cartilage [21]. These cells are often exposed to various sources of noxious stimuli, including polypeptides, cytokines, biomechanical signals, and even fragmented components of the ECM itself [21]. Impaired homeostasis leads to an increase in water content and a decrease in proteoglycan content in the ECM. This results in a weakened collagen network due to decreased synthesis of collagen type II and increased breakdown of existing collagen. Additionally, there is an increase in the rate of chondrocyte apoptosis [34]. Ultimately, this paves the way for the onset of OA, which initiates when chondrocytes fail to maintain a balance between the synthesis and degradation of ECM components [35]. 

While macrophages can phagocytize microparticles and cellular debris, the overproduction of these particles can cause significant cell stress, making it harder to dispose of them. These accumulated particles then become mediators of inflammation themselves, stimulating chondrocytes to release more catabolic enzymes [36]. Collagen and proteoglycan breakdown products can also be processed by synovial macrophages, triggering the release of TNFα, IL-1, and IL-6. These molecules can bind to chondrocyte receptors, leading to the further release of MMPs and inhibition of the synthesis of collagen type II. This sequence of events can exacerbate cartilage degeneration and create a more debilitated microenvironment [37].

## 4. PRP Versus PRF

### 4.1. PRP

PRP is defined as an autologous preparation obtained from the centrifuged peripheral blood of patients with a concentration that is 2–5 times greater than the basal concentration [38,39,40,41]. This orthobiologic product can be processed manually or with the assistance of commercial kits; however, since there are many different protocols in the literature there is a lack of consensus regarding an idealistic preparation of PRP. This also creates variance in content among PRP products and, therefore, different terminologies [42,43,44]. 

Out of the three platelet granules (alpha, delta, and lambda), the α-granules found in PRP are the most abundant [45]. The number of α-granules per thrombocyte is estimated to lie around 50–80, constituting approximately 10% of platelet volume [46]. Platelet granules are known to contain a large number of bioactive molecules which, upon activation, are released and subsequently stimulate the natural healing cascade [42,47]. The dense delta granules carry molecules such as magnesium, calcium, adenosine, serotine, and histamine, which stimulate clotting [45]. Lambda granules are often regarded as lysosomes because, much like these cellular organelles, they carry enzymes involved in protein, lipid, and carbohydrate degradation. Therefore, they are also responsible for the removal of debris and infectious agents from injured tissue [48]. The α-granule proteins, in turn, are involved in crucial biological events including inflammation, clotting, host defense, cell adhesion, and cell growth [45]. 

A treatment plan with PRP ensures accelerated neovascularization, increasing blood supply and nourishment of nearby cells. This is essential for cellular regeneration and the restoration of damaged tissue [49]. Additionally, PRP can improve other biological events including recruitment, proliferation, and differentiation of cells, contributing to the proper healing of complicated wounds and tissue injury [50]. In fact, there are multiple studies in the literature which have long shown the benefits of PRP therapy for many musculoskeletal diseases, especially knee OA. Recent randomized clinical trials (RCTs), meta-analyses, and systematic reviews [51,52,53,54,55] have yet again shown that when compared with conventional alternatives such as HA and NSAIDs, both leukocyte-poor and leukocyte-rich PRP have superior results. In the majority of the studies found, PRP had a more significant effect regarding improvements in pain and function in symptomatic knee OA with safety and efficacy. Only one of the most recent RCTs [56] showed that a single intra-articular injection of leukocyte-poor PRP with HA (Artz or HYAJOINT Plus) is effective and safe for 6 months in knee OA patients. Patients showed significant improvements in visual analogue scale (VAS) pain, Western Ontario and McMaster Universities Osteoarthritis (WOMAC) indexes, Lequesne indexes, and Single Leg Stance (SLS) tests at 1, 3, and 6 months post-intervention.

There has been much investigation aiming to unveil the regenerative mechanisms of platelet concentrates. The benefits associated with PRP therapy, for example, were long believed to be only a result of the relative abundance of growth factors and their individual biological roles (Table 2). However, further research showed that PRP also promoted important secondary effects, including inflammatory modulation [38,57,58,59,60], anticatabolic activity [61], normalization of integral autophagy [62,63], cytokine regulation [64,65,66], and pro-anabolic stimuli [67,68,69]. More importantly, it is also directly involved in fibrinolytic reactions, which are a key step in the resolution of tissue injury [70]. This is of particular biological value as the fibrinolytic system, as a whole, is necessary for the recruitment of mesenchymal stem cells (MSCs), which play an indispensable role in tissue repair [42,71]. Lastly, another beneficial effect associated with PRP therapy is mononuclear cell recruitment. Thrombin and platelet factor 4 (PF4) released by platelets promote recruitment of monocytes and their subsequent differentiation into macrophages [72,73,74]. The role of these cells has been much appreciated by researchers due to their plasticity. They have the inherent ability to switch phenotypes (polarization) and also transdifferentiate into other cell types like endothelial cells in order to display additional functions in response to biological cues in the wound microenvironment [75,76,77]. Macrophages express two major phenotypes: M1 and M2. M1 is induced by microbial agents, therefore assuming a more proinflammatory role associated with wound debridement. M2 is generally produced by type II responses, conveying anti-inflammatory properties characterized by the upregulated expression of IL-4, IL-5, IL-9, and IL-13 [76]. Macrophage polarization is mostly driven by the final stages of wound healing, as M1 macrophages trigger neutrophil apoptosis, initiating clearance [78]. Once neutrophils are phagocyted, proinflammatory cytokine production is switched off and macrophages are then allowed to undergo polarization and release TGF-β1. This molecule plays a pivotal role in regulating myofibroblast differentiation for wound contraction, thus allowing resolution of inflammation and initiation of the proliferative phase in healing [70].

It is worthy to note that while PRP provides many benefits, under certain circumstances it can cause adverse effects such as infection or injury to nerves or blood vessels and morbidity at the injection site, which may depend on the operator’s skill and the patient’s health status. Individuals with a compromised immune system or with increased predisposition to certain illnesses have a higher risk of developing infection at the injured area [79,80]. Furthermore, PRP may not always be the most superior alternative for some musculoskeletal diseases [81,82], considering variance in content among PRP products and the vast number of additional orthobiologic products commercially available.

**Table 2 gels-09-00553-t002:** Growth factors in PRP.

Reference	Name	Abbreviation	Biological Role
Teven et al., 2014 [83]	Fibroblast growth factor	FGF	Regulates cell proliferation, survival, migration, and differentiation.
Bao et al., 2009 [84]	Vascular endothelial growth factor	VEGF	Stimulates angiogenesis, macrophage and neutrophil chemotaxis, migration and mitosis of endothelial cells, and increases permeability of blood vessels.
Al-Samerria and Radovick, 2021 [85]	Insulin-like growth factor	IGF-1	Regulates cell growth and differentiation, stimulates collagen synthesis, and recruits cells from bone, endothelium, epithelium, and other tissues.
Mantel and Schmidt-Weber, 2011 [86]	Transforming growth factor-β	TGF-β	Boosts production of collagen type I, stimulates angiogenesis and chemotaxis of immune cells, and inhibits osteoclast formation and bone resorption.
Nakamura and Mizuno, 2010 [87]	Hepatocyte growth factor	HGF	Secreted by mesenchymal cells, HGF stimulates mitogenesis, cell motility, and matrix invasion.
Andrae et al., 2008 [88]	Platelet-derived growth factor	PDGF	Increases collagen expression, bone cell proliferation, chemotaxis and proliferation of fibroblasts, and macrophage activation.
Zeng and Harris, 2014 [89]	Epidermal growth factor	EGF	Stimulates proliferation and differentiation of epithelial cells, and promotes secretion of cytokines by mesenchymal and epithelial cells.

### 4.2. PRF

Fibrin is another essential component found in platelet-rich hemoderivatives, but it is often misprized and therefore deserves its fair share of credit in tissue repair. PRF, in particular, is a major source of fibrin and is acknowledged as a second generation of platelet concentrates. Although somewhat similar, PRF has a few comparable improvements over the traditional PRP, which has minor drawbacks such as blood handling as well as the addition of anticoagulants [90]. This alternative biomaterial represents a natural fibrin matrix that not only displays immunological and stimulatory properties but also holds all the hematological components that are naturally involved in the healing cascade [90]. PRF works like an autologous cicatricial matrix that is simply the result of centrifuged peripheral blood without the addition of any external agents [91]. In reality, this biograft is a fibrin-matrix polymer with a tetramolecular structure, incorporating platelets, leukocytes, cytokines, and even circulating stem cells [92]. The adhesive properties of PRF [93] can convey superior advantages as this material may last longer in target joints, thus sustaining prolonged growth factor delivery and stable cell adhesion and proliferation. 

Research indicates that at least in comparison with PRP, PRF could still contribute to the healing process without triggering intense “flare-ups” that can occasionally happen with PRP injections, especially when using leukocyte-rich PRP [10]. PRF offers some versatility in the sense that it can also be easily manipulated and used as a membrane, assisting in the closure of chronic wounds, or even as a platelet gel in conjunction with other biomaterials such as bone grafts, targeting improvements in bone tissue repair [94,95,96,97]. 

Although there are various protocols describing the preparation of platelet concentrates, the preparation of PRF is very simple with minor differences, starting with the exclusion of anticoagulants, which are known suppressors of tissue regeneration [98]. In simple terms, venipuncture is performed and blood is collected into plastic tubes, which may or may not be coated with a clot activator (i.e., glass, silica, thrombin) to accelerate coagulation. The tubes are then immediately placed in a centrifuge and only one round of centrifugation is performed, usually at 400× *g* for 10 min [99]. After centrifugation, a heterogeneous mixture is obtained: erythrocytes, which are the densest particles, are separated from the plasma and remain at the bottom layer of the suspension, whilst platelet-poor plasma (PPP) occupies the superior fraction (Figure 1C). The fibrin clot that is formed between the PPP and the erythrocyte layer is what makes PRF such an interesting orthobiologic tool. It entraps platelets, leukocytes, and growth factors, thus becoming a natural and autologous bioscaffold (Figure 2) and a rich reservoir of bioactive molecules for tissue regeneration [99,100,101,102,103]. PRF can then be easily removed and collected from the tube with forceps for further applications. 

The fundamental mechanism at play is the combination of fibrinogen with circulating thrombin as centrifugation occurs in order to form fibrin and, ultimately, the fibrin clot [90]. Platelet activation and fibrin polymerization are two important events that take place. Platelets are activated immediately upon contact with the walls of the tubes, leading to the formation of the dense fibrin network which gives the PRF clot its typical characteristics [99]. Time is of the essence; when working with PRF, the operator must be able to quickly collect blood and centrifuge it immediately in order to avoid premature coagulation. Ideally, this should be performed within 2 min and 30 s [99]. Any prolongation during this stage causes diffuse fibrin polymerization, making the obtained PRF sample unsuitable for clinical use [99,104]. Despite these observations and “technical complications”, it is interesting to note that an injectable version of PRF (i-PRF) also exists, offering more feasibility in treatment alternatives [98]. Injectable PRF is typically prepared in vacuum tubes with no additives (Figure 1) in order to delay coagulation. This can provide physicians with sufficient time to process the orthobiologic sample as needed and make an intra-articular application. It is worthy to note that PRP alone is known to be a quite diffusible biostimulator, which can sometimes hinder its therapeutic value depending on where it is being injected [105]. Fibrin, however, is capable of retaining the shape of an articulation for about 5 min after injection, holding HA in the injected sites [106]. Therefore, the association of HA, i-PRF, and PRP (power mix), as shown in Figure 3, may produce enhanced biological effects, allowing them to remain for an extended amount of time in the target joint.

Clinical studies and other previous investigations (Table 3) have revealed positive effects regarding the administration of PRF either alone or in association with other products for OA of different joints. A recent pilot study [105] comparing leukocyte- and platelet-rich fibrin (L-PRF) versus PRP + HA in the treatment of patients affected by unilateral knee degenerative OA revealed that the fibrin counterpart promotes better pain control and longer pain relief in the short and medium terms. 

Recent findings from a double-blind clinical study [107] revealed that the combination of high molecular weight (6,000,000 Daltons) HA with plasma fibrinogen conveys positive effects on knee OA symptoms for all assessed parameters, marked by a significant reduction in OA-associated pain. The fibrin-HA formulations used in this clinical trial (RegenoGel and RegenoGel-OSP) demonstrated significant improvements in VAS scores compared with the placebo at three months after the first injection, and sustained these improvements for another three months after the second injection. These results indicate that fibrin and HA conjugates are safe and potentially effective for at least six months in the treatment of knee OA (KOA) symptoms. Indeed, combining HA with fibrin can enhance biological effects. Scanning electron microscopy of HA + fibrin composites shows that this association promotes more robust fibrin polymerization from solid to porous structures [106]. This can further contribute to cell survival, attachment, migration, and proliferation [108,109].

In similar fashion, a recent prospective study [110] aimed at evaluating a 36-month survival analysis of conservative treatment using PRP enhanced with i-PRF in 368 knee OA patients. The results showed that the overall survival rate of knees that did not require surgical intervention during the 36-month follow-up was 80.18%, which is considered satisfactory. This reveals a potentiating property for i-PRF, which in this case was able to enhance regenerative effects upon being combined with PRP.

In cranio-maxillofacial applications, PRF has also demonstrated promising results in temporomandibular joint (TMJ) disorders, such as TMJ-OA. A recent RCT [111] assessed the treatment outcomes of intra-articular (IA) delivery of i-PRF after arthrocentesis in patients with TMJ-OA. In comparison with the control group (arthrocentesis alone), i-PRF demonstrated superior results in terms of pain reduction and improvements in functional jaw movements. Another similar clinical study [112] evaluated the effects of IA i-PRF treatment in patients suffering from unilateral click due to TMJ disorders. The clicking completely disappeared in 70% of the patients in just 1 week after the first injection, and then in all patients 1 week after the second injection. Taken together, this evidence indicates PRF’s efficacy in the management of TMJ disorders as well, demonstrating satisfactory results in terms of pain relief and functional gains.

There are many molecular mechanisms underpinning the efficacy of platelet- and fibrin-rich products; however, particular importance should be given to the fibrinolytic systems, which are discussed in the subsequent sections of this manuscript.

**Table 3 gels-09-00553-t003:** Summary of PRF studies.

Reference	Disorder	Treatment	Injection	Outcome
Di Nicola, 2020 [105]	Unilateral knee OA KL grades II–III	PRP + HA or L-PRF alone	Single intra-articular knee injection	L-PRF is superior in pain improvement in just 30 days
Jang et al., 2013 [106]	Knee OA KL grades II, III, and IV	(Fibrin + HA) RegenoGel or RegenoGel-OSP, or saline (placebo)	Intra-articular knee injection	Both fibrin + HA formulations are superior to placebo 3 and 6 months post-treatment in terms of VAS pain score improvements
Kandel et al., 2020 [107]	Articular cartilage defect in rabbits	Fibrin + HA, bone marrow concentrate + fibrin + HA, control	Injection into the cartilage defect area in the knee	Complete regeneration with smooth surface in both experimental groups
Cheeva-akrapan and Turajane, 2023 [110]	Knee OA KL grades I–IV	PRP + i-PRF	Intra-articular knee injections—supine position, knee flexed at 90 degrees; anteromedial joint space	80.18% of knees did not require surgical intervention during the 36-month follow-up
Işık, 2022 [111]	Temporomandibular joint osteoarthritis	i-PRF; arthrocentesis (control)	Mandibular intra-articular PRF injection	Pain decrease and functional improvements in jaw movement were superior in the i-PRF group
Manafikhi, Ataya, and Heshmeh, 2022 [112]	Temporomandibular joint disorder (unilateral click)	i-PRF	Superior joint space of the TMJ with the internal disorder	Clicking disappeared in 70% of patients in 1 week after the first injection Clicking disappeared in all patients 1 week after the second injection (1-week interval)

### 4.3. Hyaluronic Acid

Hyaluronic acid is an anionic, nonsulfated glycosaminoglycan abundantly found in multiple organ systems [113]. 

HA products with higher molecular weights normally sustain anti-inflammatory effects because they regulate immune cell recruitment. On the other hand, formulations with lower molecular weights have been reported to promote angiogenesis and tissue remodeling in wound healing; however, they may also display a more proinflammatory effect in specific cell types, especially chondrocytes [114,115].

Low-molecular weight (LMW) (500,000 to 730,000 Daltons) HA binds to cell surface receptors less efficiently, which in turn promotes weak HA biosynthesis. In fact, this formulation is not very beneficial and is rather associated with inflammation. To elaborate, Tanimoto et al. [116] evaluated the effects of the proinflammatory cytokines TNF-α and IL-1β on leporine HA-synthetase (HAS) mRNA expression. Under inflammatory conditions typically seen in OA, these cytokines increase HAS mRNA expression, contributing to fragmentation and overaccumulation of HA, which is detrimental to cellular processes.

Medium-molecular weight (MMW) HA allows for stronger binding, stimulating a higher number of HA receptors, and thus enhancing endogenous HA production. It is worthwhile to note that extremely large molecules present in high-molecular weight (HMW) HA products may not always be convenient [117]. These molecules still bind to HA receptors; however, their large domains can limit the number of free binding sites on the cell surface, which logically implies a less efficient stimulation of HA biosynthesis [117]. 

HA is also an essential component of articular cartilage, in which it exists as a protective layer surrounding chondrocytes. It acts as a lubricant for tendons and joints, reducing extracellular matrix (ECM) degradation via inhibition of matrix metalloproteinase (MMP) synthesis. Its anti-inflammatory effects attenuate the activity of tumor necrosis factor-alpha (TNF-α) and interleukin-1 (IL-1), two major proinflammatory mediators. By itself, HA can be a potent agent in the management of OA, especially of the knee. Its benefits for orthopedic conditions have been well documented in the literature for decades. More recently, systematic reviews have yet again shown that IA applications of HA are safe and cost-effective as they reduce pain and improve knee function in comparison with conservative treatments, including NSAIDs, corticosteroids, and analgesics [118,119]. IA-HA is regarded as a minimally invasive interventional strategy and there are no records of major systemic adverse events [120]. This approach has shown beneficial effects in vitro. IA-HA has been shown to not only reduce chondrocyte apoptosis but also increase its proliferation [121]. In humans, it is best to use formulations with medium- (800,000–2,000,000 Daltons) to high-molecular weight (MW) HA in order to closely emulate the conditions and biological properties of HA naturally produced in the body. Also, it is important to use HA derived from biological synthesis, to avoid undesired side effects [122].

The molecular mechanisms of this modality are attributed to the ability of HA to bind to cluster of differentiation 44 (CD44) receptors. This blocks the expression of IL-1β, therefore downregulating the production of MMPs 1, 2, 3, 9, and 13 [123,124,125], bypassing the activity of catabolic enzymes in musculoskeletal structures [126]. After binding to its receptor, HA triggers intracellular signaling pathways associated with the proliferation, differentiation, migration, and degradation of HA itself [127]. CD44 is the most widely studied HA receptor because it is expressed in nearly all human cell types. Affinity between CD44 and HA is a crucial factor that dictates the potential of HA as a signaling molecule. However, this also depends on HA concentration and MW, as well as glycosylation of extracellular domains and phosphorylation of serine [128]. CD44 can form clusters with HMW HA polymers, allowing interaction with growth factors, ECM proteins, MMPs, and cytokines [129]. Another important HA surface receptor is CD168. It is expressed in multiple cells and controls migration by interacting with skeletal proteins, especially in the healing cascade [128]. 

### 4.4. The Role of Fibrin in Regeneration

#### Fibrinolytic Reactions

Platelets contain various factors related to the fibrinolytic system that can either increase or decrease fibrinolytic reactions. The extent of contribution of different hematological components and the role of platelets in the degradation of fibrin are still debated in the scientific community [130]. While previous research tends to focus solely on platelets, other components such as coagulation factors and the entire fibrinolytic system have also been found to play an important role in efficient tissue repair [70]. In simple terms, fibrinolysis is a complex process that requires the activation of specific plasma proteins to promote the degradation of fibrin [70]. Previous studies have acknowledged the importance of fibrinolytic reactions, suggesting that fibrin degradation products (FDPs) could potentially drive tissue repair, preceding the sequence of events from fibrin deposition and removal all the way to angiogenesis, which is vital for wound healing [131]. 

When injury occurs, the formation of a clot is essential to protect tissues from blood loss and microbial invasion. However, the clot also serves as a provisional matrix that allows efficient cell migration during repair [132,133]. This clot is made up of an aggregation of platelets embedded in a mesh of cross-linked fibrin fibers resulting from the cleavage of fibrinogen by serine proteases. This reaction triggers the polymerization of fibrin monomers, which is the primary event in the formation of a blood clot [70]. The clot can also act as a storage site for cytokines and growth factors that are released upon degranulation of activated platelets [70,134]. 

The fibrinolytic system, which is tightly regulated by plasmin, plays an important role in cell migration, regulation of growth factor availability, and other protease systems involved in tissue inflammation and regeneration [135,136]. The two major components involved in fibrinolytic reactions are urokinase plasminogen activator receptor (uPAR) and plasminogen activator inhibitor-1 (PAI-1). Both are expressed in mesenchymal stem cells (MSCs), which are specialized cells that are necessary for successful wound healing [135]. 

Previous studies have suggested that uPAR, in particular, plays a key role in MSC mobilization, similar to what occurs in hematopoietic stem cell (HSC) mobilization. Vallabhaneni and colleagues demonstrated that the administration of granulocyte colony-stimulating factor in uPAR-deficient mice resulted in the failure of MSC mobilization [137], emphasizing the supportive role of fibrinolytic systems in stem cell migration. Further research has shown that the glycosylphosphatidylinositol-anchored uPA receptor activates specific intracellular signaling pathways, such as prosurvival phosphatidylinositol4,5-bisphosphate 3-kinase/Akt and ERK1/2 signaling, and focal adhesion kinase (FAK), to regulate adhesion, migration, proliferation, and differentiation [138,139].

In the context of wound healing, fibrinolytic factors play a crucial role. Studies on mice have shown that plasminogen deficiency causes a severe delay in wound healing events, suggesting the indispensable role of plasmin in this process [140]. In humans, loss of plasmin also leads to complications in wound healing. Impaired blood flow can significantly stagnate tissue regeneration, which would also explain why these regenerative processes are much more challenging in diabetic patients, for example [141,142]. MSCs are recruited to the wound site to accelerate the healing process [143]; under normal conditions, they express uPA, uPAR, and PAI-1, which are important for wound healing [144]. The proteins uPAR and PAI-1 are regulated by hypoxia inducible factor α (HIF-1α) in MSCs [145], which is beneficial because activating HIF-1α increases the production of FGF-2 and HGF; HIF-2α in turn increases the production of VEGF-A [146]. These molecules collectively aid in wound healing, as shown in Table 2. Additionally, HGF also enhances recruitment of MSCs into the wound site in a synergistic manner [147]. 

It is important to remember that wounds with poor blood flow and low oxygen levels do significantly hinder the healing cascade [148]. Although MSCs typically inhabit relatively hypoxic tissues (bone marrow), transplanted MSCs may not fare so well in wounds with such adverse conditions, which can limit their healing potential [148]. The way in which MSCs attach and survive in hypoxic conditions depends on their production of fibrinolytic factors. PAI-1 competes with uPAR and integrins to bind to vitronectin, which results in the inhibition of cell adhesion, migration, and proliferation [149]. However, when PAI-1 interacts with PAs, the affinity of PAI-1 for vitronectin is lost, which allows for cell migration [135]. This process determines whether the MSCs adhere to a surface or migrate, and it also affects their survival.

## 5. Limitations and Conclusions

Fibrin-rich orthobiologic products such as autologous platelet concentrates have long been studied and appreciated for their positive effects on numerous conditions. The application of PRP and its derivatives, such as PRF, has produced optimistic results in both clinical and laboratory scenarios regarding musculoskeletal disorders, especially OA. There are, however, some limitations. In some investigations, PRP may not always display major differences that are of significant benefit to patients when compared with control/placebo or other orthobiologics. This may also be the case for PRF, especially when taking into consideration that it must be worked with quickly.

Although PRP and PRF have particularities of their own, they also share essential similarities. The relative abundance of platelets, peripheral blood cells, and molecular components in these orthobiologic products provokes many biological events. Examples include inflammatory modulation, enhanced neovascularization, and delivery of pro-anabolic stimuli, which regulates the recruitment, proliferation, and differentiation of cells. Furthermore, the fibrinolytic system has also been demonstrated to orchestrate the recruitment of cells like leukocytes and MSCs, and regulation of proteolytic activity in areas of tissue regeneration, such as bone, cartilage, and muscle. Therefore, it is an important component in musculoskeletal regenerative medicine.

PRP is often regarded as a potent signaling agent, but it is quite diffusible; PRF offers a durable scaffolding effect, favoring cell activity. However, combining HA with fibrin enhances scaffolding properties, as this association promotes more robust fibrin polymerization, aiding cell survival, attachment, migration, and proliferation. Therefore, the autologous administration of the “power mix” may prove to be feasible in regenerative medicine, being cost-effective and low-risk. In cases where tissue repair is often challenging, such as OA, the application of the power mix may prove to be a suitable management strategy. One way to ascertain the quality of PRP is to submit an aliquot to a hematology cell counter in order to verify the concentration of platelets, which should be between two and five times greater than the basal value. In terms of the fibrin matrix, the gel-like mixture should ideally be firm and robust.

Despite the growing number of platelet- and fibrin-rich derivatives for the treatment of orthopedic diseases, future investigations are still needed in order to further comprehend the factors that contribute to musculoskeletal tissue repair. 

## 6. Future Direction

The administration of the autologous “power mix” (PRP + PRF + HA) as shown in Figure 3 may be a viable solution in regenerative medicine for musculoskeletal conditions, as it is cost-effective and low-risk. The fibrin in the mix serves as a natural adhesive matrix that aids in the delivery of growth factors and bioactive molecules to the cells. When combined with additional PRP components, it may enhance the regenerative effects because the fibrin matrix itself has a strong affinity for platelets. This allows platelets to efficiently degranulate and release their contents for an extended amount of time in the local wound microenvironment. This in turn effectively modulates inflammation and recruitment of additional cells, therefore facilitating the progression of tissue repair.

## Figures and Tables

**Figure 1 gels-09-00553-f001:**
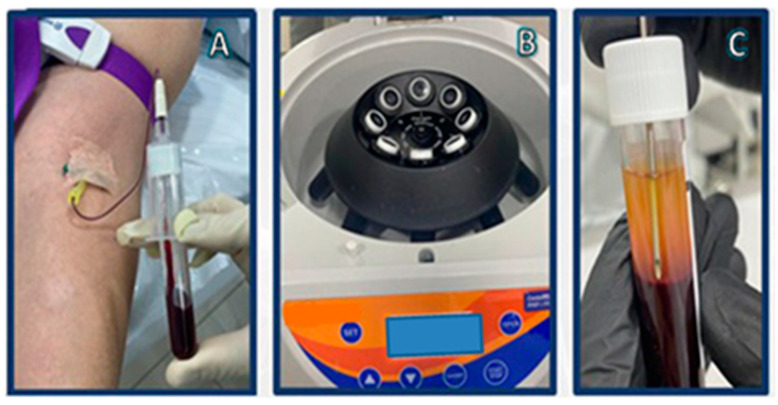
i-PRF preparation: (**A**) vacuum-extraction blood specimen collection; (**B**) centrifugation; (**C**) i-PRF aspiration.

**Figure 2 gels-09-00553-f002:**
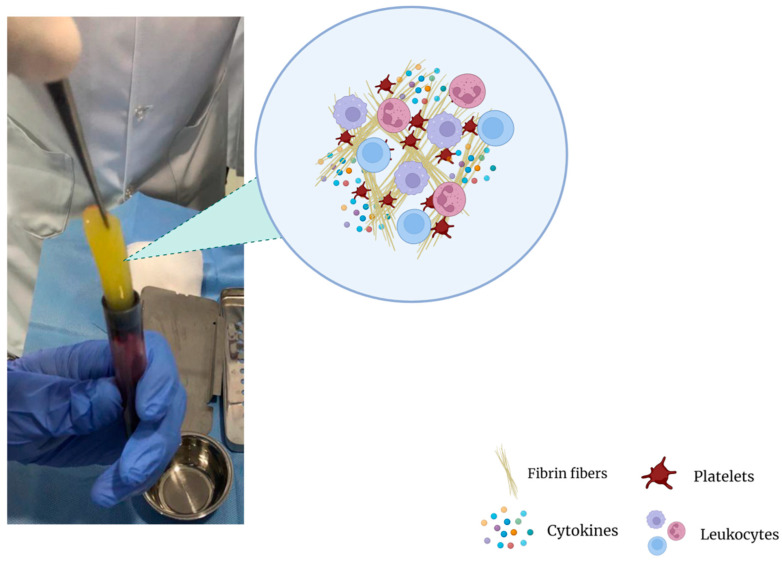
Fibrin clot matrix.

**Figure 3 gels-09-00553-f003:**
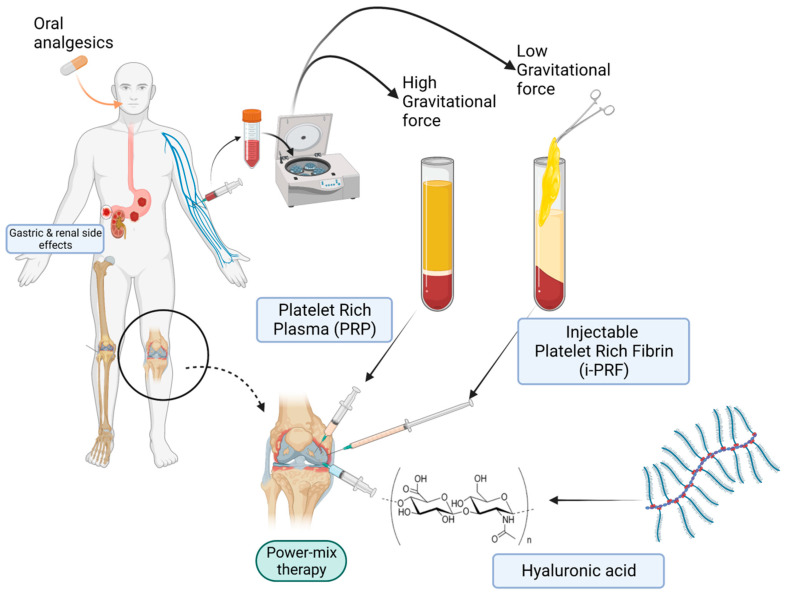
Power-mix (PRF-PRP-HA matrix) therapy versus conventional analgesics.

**Table 1 gels-09-00553-t001:** Osteoarthritis classification based on the Kellgren–Lawrence scale.

Osteoarthritis Grade	Observation
Grade 0 (normal)	No radiological findings
Grade I (doubtful)	Possible signs of osteophytic lipping and narrowing of joint space
Grade II (mild)	Definite osteophytes and possible joint space narrowing
Grade III (moderate)	Definite joint space narrowing and multiple osteophytes
Grade IV (severe)	Large osteophytes, prominent demarcation of narrowed joint space, severe sclerosis, and expressive deformity of bone contour

## Data Availability

Not applicable.

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
