# Peer review of "Platelet-Rich Plasma Power-Mix Gel (ppm)—An Orthobiologic Optimization Protocol Rich in Growth Factors and Fibrin"

_gels, 2023, doi:10.3390/gels9070553_

Round 1
Reviewer 1 Report
Please see attached

The document is quite easy to comprehend, there are some structural flow issues where the topics or sentences do not connect well. Some content appears out of order where it may make sense to have it earlier in the review.
Reviewer 2 Report
Please see attached document

Minor editing of English language required
Round 2
Reviewer 1 Report
The authors have taken into consideration the suggested edits and have produced a greater quality manuscript.
Author Response
Dear reviewer 1,
Thank you once again for revising our manuscript and making valuable suggestions.
We are glad to know our attempts to improve the manuscript matched your comments and suggestions.
Reviewer 2 Report
Please see attached document

Minor editing of English language required
Author Response
Dear reviewer, we would like to thank you once again for revising our modifications.
According to your suggestions we have now added a small paragraph containing 4 additional references regarding limitations of PRP. However, these were added right at the end of section 4.1 (PRP) and not at the conclusions section.
Thank you very much for all your comments and suggestions which highly contributed to the improvement of our manuscript.